# Deciphering the Pathophysiological Mechanisms Underpinning Myoclonus Dystonia Using Pluripotent Stem Cell-Derived Cellular Models

**DOI:** 10.3390/cells13181520

**Published:** 2024-09-10

**Authors:** Zongze Li, Laura Abram, Kathryn J. Peall

**Affiliations:** 1Neuroscience and Mental Health Innovation Institute, Cardiff University, Hadyn Ellis Building, Maindy Road, Cardiff CF24 4HQ, UK; liz55@cardiff.ac.uk (Z.L.); abraml@cardiff.ac.uk (L.A.); 2Division of Psychological Medicine and Clinical Neurosciences, Cardiff University, Cardiff CF24 4HQ, UK

**Keywords:** dystonia, Myoclonus Dystonia, *SGCE*, epsilon-sarcoglycan, pluripotent stem cell, disease modelling, pathogenesis

## Abstract

Dystonia is a movement disorder with an estimated prevalence of 1.2% and is characterised by involuntary muscle contractions leading to abnormal postures and pain. Only symptomatic treatments are available with no disease-modifying or curative therapy, in large part due to the limited understanding of the underlying pathophysiology. However, the inherited monogenic forms of dystonia provide an opportunity for the development of disease models to examine these mechanisms. Myoclonus Dystonia, caused by *SGCE* mutations encoding the ε-sarcoglycan protein, represents one of now >50 monogenic forms. Previous research has implicated the involvement of the basal ganglia–cerebello-thalamo-cortical circuit in dystonia pathogenesis, but further work is needed to understand the specific molecular and cellular mechanisms. Pluripotent stem cell technology enables a patient-derived disease modelling platform harbouring disease-causing mutations. In this review, we discuss the current understanding of the aetiology of Myoclonus Dystonia, recent advances in producing distinct neuronal types from pluripotent stem cells, and their application in modelling Myoclonus Dystonia in vitro. Future research employing pluripotent stem cell-derived cellular models is crucial to elucidate how distinct neuronal types may contribute to dystonia and how disruption to neuronal function can give rise to dystonic disorders.

## 1. Introduction

Dystonia is a heterogeneous spectrum of rare movement disorders characterised by the loss of co-ordinated contraction of antagonistic muscle groups. Based on aetiology, dystonia can be categorised as inherited, acquired, or idiopathic. More than 50 genes have now been identified as causative for the inherited forms of dystonia, with examples including *ANO3*, *GNAL*, *GNAO1*, *THAP1*, and *TOR1A.* Myoclonus Dystonia (MD) is another such inherited form of dystonia, caused by mutations in the *SGCE* gene that encodes the ε-sarcoglycan protein, and is inherited in an autosomal dominant fashion but with reduced penetrance owing to maternal imprinting. MD is characterised by the combined presentation of dystonia and myoclonus (brief, involuntary movements) that are typically alcohol-responsive [1]. The onset of the motor symptoms is typically in childhood with three distinct patterns of symptomatic evolution [1,2]. Co-morbid psychiatric symptoms are also well recognised in MD, most notably, generalised anxiety disorder, specific and social phobias, obsessive–compulsive disorder, alcohol dependence, and depression [3,4,5,6]. However, the pathogenesis underpinning MD and the wider spectrum of dystonic disorders remains unknown. While previous research has employed a variety of approaches, including brain imaging, human in vivo electrophysiological studies, and animal models, recent advances in pluripotent stem cells (PSCs) and subsequent neuronal differentiation have enabled the use of human PSCs, particularly induced pluripotent stem cells (iPSCs) derived from patients, to study the molecular and cellular mechanisms of dystonia [7,8,9,10,11,12,13,14]. Here, we review the current understanding of the aetiology of MD and recent advances in generating distinct neuronal subtypes that are considered relevant to MD pathogenesis coupled with future avenues to aid the pathophysiological understanding and novel therapeutic target identification.

## 2. Clinical Phenotype

The onset of the motor features of MD is typically in the first two decades of life, with a median age at onset of 3 years [1,15]. MD involves two predominant motor phenotypes: myoclonus and dystonia [1]. Myoclonus is characterised by sudden and “lightning-like” jerks, which in MD primarily affect the trunk and upper limbs, are most pronounced with posture or activity, and are typically alcohol-responsive [2,16,17,18]. The dystonic component of MD typically presents in focal or segmental forms, predominantly affecting the neck (cervical dystonia) or the upper limbs, and in the form of writer’s cramp [1] (Figure 1A). Longitudinal studies have observed three main patterns of motor symptom evolution: (i) early childhood onset upper body myoclonus and dystonia, (ii) later childhood onset upper body myoclonus and dystonia with evident cervical involvement, and (iii) early childhood onset lower limb dystonia, progressing to more pronounced myoclonus and upper body involvement [2] (Figure 1B). Multiple case reports and case series have described the symptomatic benefit from a range of oral medical therapies, although zonisamide remains the only one in which benefit has been demonstrated in the context of a randomised clinical trial (Table 1). Surgical intervention, in the form of deep brain stimulation (DBS) to the globus pallidus internus (GPi) has also been shown to lead to improvements in both myoclonus and dystonia across adult and paediatric patients.

A broader non-motor phenotype is also observed in MD with a predominant focus on psychiatric symptomatology. A case control analysis has demonstrated an excess of anxiety, depression, obsessive–compulsive disorder, and social phobia amongst those harbouring *SGCE* mutations, compared to unaffected controls [3,6], with a longitudinal analysis demonstrating a progression of these symptoms in adult life [28]. In addition, the assessment of cognitive function has shown higher level deficits, including impairments to executive function [29].

## 3. SGCE and the ε-Sarcoglycan Protein

Previous work has identified pathogenic *SGCE* mutations in giving rise to MD [4,5]. The *SGCE* gene is maternally imprinted, with the CpG islands in the promotor and exon 1 methylated on the maternal allele, leading to an almost exclusive expression of the paternal allele [5] and reduced penetrance of the clinical phenotype when maternally inherited [30]. Multiple mutation types are recognised to cause MD including point mutations, short insertions and deletions, and contiguous gene deletions [4,31,32,33,34]. Nonsense mutant transcripts are degraded via nonsense-mediated mRNA decay, while missense mutant transcripts are thought to produce a mutant protein that is mis-localised in the neuronal soma and degraded via the proteasome pathway [35].

The *SGCE* gene encodes the ε-sarcoglycan protein, a single-pass transmembrane protein, which is part of the sarcoglycan family (other members include α-, β-, γ-, δ-, and ζ-sarcoglycan). Sarcoglycan proteins form heterotetrametric complexes and participate in the formation of the dystrophin-associated glycoprotein complex (DGC), which plays a central role in the cell mechano-transduction and homeostatic signalling in muscle cells [36]. In striated muscles, the sarcoglycan complex consists of α-, β-, γ-, and δ-sarcoglycans [37,38,39], while brain sarcoglycan heterotetramers are formed by β-, δ-, ε-, and ζ-sarcoglycans [40] (Figure 2). Brain sarcoglycan heterotetramers may also contribute to brain DGC-like protein complexes, which potentially participate in the formation and function of the GABAergic synapses (Figure 2) and the function of the blood–brain barrier and astrocytes by regulating aquaporin-4 localisation and cell adhesion [40,41,42,43,44,45,46,47,48,49,50].

Although the *SGCE* gene is ubiquitously expressed in both the muscle and non-muscle tissues as early as embryonic day 12 in mice, the *SGCE* gene shows region-specific isoform expression and alternative exon splicing. Specifically, exon 2 is ubiquitously expressed, exon 8 is almost exclusively detected outside the brain, and exon 11b is specifically enriched in the brain [51,52]. Within the brain, *SGCE* expression is detected in all brain regions but is most highly expressed in the cerebral cortex, cerebellar Purkinje cell layer, hippocampus, and basal ganglia [52,53]. In different cell types, single-cell RNA-sequencing detected the highest *SGCE* transcript expression in oligodendrocyte precursors, oligodendrocytes, and excitatory neurons [54]. Overall, the expression profile of *SGCE* and involvement in the DGC implicates the cell types and brain regions affected in MD, namely the basal ganglia, cerebral cortex, and cerebellum.

## 4. Pathophysiology: A Network View

Disruption to the basal ganglia–cerebello-thalamo-cortical circuit has been recognised as a common feature across dystonic disorders [55,56]. Therefore, although this review focuses on MD, this section will discuss the role of these individual regions in the pathogenesis of dystonia (Figure 3).

### 4.1. Basal Ganglia

The basal ganglia are composed of two neuronal pathways, direct (mediated by Dopamine Receptor D1 (DRD1)-expressing striatal medium spiny neurons (MSNs)) and indirect (mediated by Dopamine Receptor D2 (DRD2)-expressing MSNs), which converge on the GABAergic neurons in the globus pallidus internus (GPi) and substantia nigra pars reticulata, projecting to the ventral thalamus [57]. Direct pathway stimulation leads to cortical activation and movement facilitation, while excitation of the indirect pathway results in cortical inhibition and reduced movement [57]. It has been hypothesised that the imbalance between the direct and indirect pathways and the abnormal spatial and temporal patterns of basal ganglia activity lead to hypo-activity in the GPi, resulting in the increased excitability of the motor cortex in dystonia [58]. Supporting the role of the basal ganglia in dystonia, globus pallidus neurons exhibited abnormal burst activities in MD during intraoperative single-unit recordings [59,60], as well as excessive synchronised low-frequency oscillation during local field potential recordings [60,61,62,63,64]. Moreover, DBS of the GPi is effective in the treatment of MD [65,66,67], suggesting that hypo-activity of the basal ganglia output nuclei play a role in dystonia.

The substantia nigra pars compacta (SNpc) is also considered to play a role in dystonia, with the main projection neurons—midbrain dopaminergic (mDA) neurons—innervating MSNs as part of the basal ganglia network [68]. Dopaminergic transmission from the SNpc modulates striatal activity by stimulating DRD1-expressing MSNs in the direct pathway and inhibiting DRD2-expressing MSNs in the indirect pathway [57]. The disruption to dopaminergic neurotransmission has been suggested to play a role in MD, with reduced DRD2 expression, higher levels of dopamine and its metabolites, increased release of dopamine in response to amphetamine in *Sgce* knock-out mice [69,70], and reduced DRD2 availability reported during brain imaging of patients diagnosed with MD [71].

### 4.2. Cerebellum

The cerebellum has increasingly been considered a key player in the pathogenesis of dystonia, with abnormalities of eyeblink classical conditioning in individuals with primary focal dystonia and MD [1,72] and increased metabolic activity in the cerebellar network of *TOR1A* mutation-positive dystonia and MD during human brain imaging studies [73,74]. Disrupting cerebellar function and transmission using a kainic acid microinjection, blocking the ATPase Na^+^/K^+^ Transporting Subunit Alpha 3 in Purkinje cells, and the selective abolishment of glutamatergic olivocerebellar synaptic input to Purkinje cells can cause dystonia-like phenotypes in rodent models [58,75,76,77]. In *Sgce* knock-out mice, cerebellar Purkinje cells have also been shown to have abnormal nuclear envelope structures [78], while *Sgce* knock-down models have altered the neuronal firing of Purkinje cells and deep cerebellar neurons coupled with dystonia-like motor deficits [79].

### 4.3. Cerebral Cortex

Multiple levels of evidence support a key role for the cerebral cortex in dystonia pathogenesis. Firstly, brain imaging studies have demonstrated changes to network activity and connectivity and increased metabolic activity in the pre-supplementary motor area and parietal association cortex, as well as reduced fractional anisotropy of the sensorimotor white matter [73,80,81,82]. Secondly, neurophysiological studies have implicated reduced inhibitory transmission, with reduced levels of GABA type A receptor-mediated short-interval intracortical inhibition in the primary motor cortex following paired-pulse transcranial magnetic stimulation (TMS) of motor manifesting and non-manifesting *TOR1A* mutation carriers [83,84]. Similar disruption to TMS-induced inhibition, albeit more subtle, has also been reported in patients diagnosed with MD [85,86,87]. Finally, altered synaptic plasticity has been observed with increased TMS-induced long-term potentiation in patients with writer’s cramp and *TOR1A* mutations, suggesting a hyperexcitable corticospinal pathway [88,89]. Moreover, the latent and progressive manifestation of the beneficial effect of GPi DBS treatment for primary dystonia suggests a gradual reversal and re-establishment of synaptic plasticity, which might be disrupted in dystonia [90,91]. Furthermore, in animal models of dystonia, an increase in long-term potentiation and the loss of long-term depression have been reported at corticostriatal synapses of mutant human *tor1a* knock-in mice [92,93].

## 5. Neuronal Differentiation from PSCs

Although animal models, human imaging, and electrophysiological studies have provided some understanding of MD, the precise underlying molecular and cellular mechanisms remain largely unknown. Existing murine *Sgce* knock-out models of MD remain limited in their applicability owing to their lack of recapitulation of the dystonia motor phenotype [69,70,94], while models involving the conditional knock-down of *Sgce* expression fail to reflect the potential developmental impact of *SGCE* mutations [79,94]. PSC-derived cellular models offer a valuable, alternative platform by which to examine the cellular pathogenesis and neuronal developmental impact of pathogenic mutations in dystonia-causing genes [56]. PSCs may be derived from the inner cell mass of preimplantation embryos, known as embryonic stem cells, or reprogrammed in vitro from somatic cells (such as skin fibroblasts and platelets)—iPSCs [95,96] (Figure 4). Due to their capability for self-renewal, versatility in differentiating into various cell types, ease of genetic and experimental manipulation, as well as the opportunity to study the effects of genetic mutations in disease-relevant genetic backgrounds, PSC-derived cellular models have been widely used in disease modelling and drug discovery [97]. A crucial step in this approach is the selection of relevant cell types to differentiate these cells, typically undertaken either through molecular reprogramming with the overexpression of transcription factors (Figure 4), or a step-wise paradigm employing combinations of growth factors and small molecule drugs for manipulating cell signalling pathways (Figure 4).

### 5.1. Neuron Differentiation by Directly Reprogramming PSCs

PSCs can be directly reprogrammed into neurons through the transgenic expression of the key transcription factors critical for neurogenesis and subtype specification [98]. Neurons with a cortical glutamatergic identity can be directly generated by inducing *NEUROG2* expression [99], while combining *NEUROG2* induction with dual SMAD inhibition during neural induction enhances the efficiency of neuronal differentiation [100,101] and can be coupled with different patterning paradigms to generate neurons with distinct regional identities (Figure 4). For example, the co-expression of *EMX1* and *NEUROG2* generates glutamatergic neurons, *ASCL1*, *DLX2*, and *LHX6*, coupled with microRNAs (*microRNA-9/9** and *microRNA-124*), generates functional GABAergic neurons resembling medial ganglionic eminence (MGE)-derived cortical interneurons, and the co-expression of *ASCL1*, *LMX1B*, and *NURR1* produces functional mDA neurons (Figure 4). This direct reprogramming strategy offers significant advantages as it is less technically demanding and time-consuming compared to the step-wise differentiation approach [98], although it bypasses the neural progenitor phase limiting investigation of the developmental defects underpinning human disease [98]. Moreover, direct reprogramming relies on a limited set of transcription factors crucial to neuronal development and subtype specification, which might not fully recapitulate the full spectrum of transcriptional and epigenetic regulation that occurs during normal neuronal development [98,102]. Recent integrative omics studies have identified a broad gene regulatory network beyond NEUROG2 essential for neuronal differentiation, underscoring this complex regulatory landscape, potentially lost during direct reprogramming.

### 5.2. Step-Wise Differentiation of Neurons

Using a step-wise differentiation strategy, PSCs are first directed towards neural progenitors (Figure 4), characterised by the expression of SOX1, SOX2, and NESTIN [103,104,105], with dual SMAD inhibition representing the most efficient approach to neural induction [105,106,107,108,109]. Studies indicate that neural progenitors generated via dual SMAD inhibition predominantly exhibit a dorsal forebrain identity, expressing PAX6, OTX1, OTX2, FOXG1, and EMX2, and form neural rosettes [106,110]. These progenitors can sequentially develop into glutamatergic neurons resembling those in different cortical layers, mimicking the ‘inside-out’ fashion of cortical projection neuron development [111,112]. However, neural progenitors with other specific regional identities can be produced using a combination of morphogens and small molecules with finely tuned temporal and dosage control [113]. Given the diversity of neuronal subtypes likely involved in MD, the next section will focus on those implicated in dystonia pathogenesis, specifically cortical GABAergic interneurons, MSNs, mDA neurons, and cerebellar Purkinje cells (Figure 4).

#### 5.2.1. Differentiating Cortical GABAergic Interneurons

Cortical GABAergic interneurons are a diverse population with various morphologic, transcriptomic, neurochemical, and electrophysiological profiles [109]. They can be broadly classified into PV+, SST+, and HTR3A+ subpopulations, accounting for the majority of the cortical GABAergic interneuronal population [110]. Cortical GABAergic interneurons primarily develop from two main embryonic regions: the MGE, which gives rise to SST+ and PV+ neurons, and the caudal ganglionic eminence, producing HTR3A+ neurons [111]. Studies have primarily focused on the production of MGE-derived cortical GABAergic interneurons due to their relevance across multiple brain disorders [112]. Current differentiation protocols typically involve the activation of Sonic hedgehog (SHH) signalling and inhibition of WNT signalling to produce NKX2.1+/FOXG1+ MGE progenitors [113,114] (Figure 3). More recent refinements have improved the overall efficiency with early WNT signalling inhibition (XAV939) followed by SHH signalling activation (with recombinant SHH and a small molecule agonist, purmorphamine) from days 10 to 18 [115] (Figure 4). By contrast, other studies that only employed SHH signalling activation or initiated WNT inhibition and SHH activation from day 0 also achieved a similar efficiency of MGE patterning [116,117,118]. More recently, Hunt et al. further defined a differentiation protocol for the MGE lineage by tuning the crosstalk of WNT and SHH signalling [114]. These protocols have successfully generated SST+ and PV+ GABAergic interneurons with mature electrophysiological characteristics, which are capable of integrating into neuronal networks when transplanted into rodent brains [115,116,117,119], including a highly pure population of pallial MGE-type GABAergic interneurons currently being tested for cell therapy [115]. Despite these advancements, challenges remain in generating the PV+ subtype efficiently from PSCs, limiting the ability of PSC-derived GABAergic interneurons to fully recapitulate the diversity found in vivo [120].

#### 5.2.2. Differentiating Medium Spiny Neurons

GABAergic MSNs, originating from the lateral ganglionic eminence (LGE), are the primary projection neurons of the striatum [116,117]. The patterning of LGE identity is influenced by the antagonistic signals—ventralising SHH signalling and dorsalising bone morphogenetic protein (BMP) signalling [118]—employed to generate LGE-derived MSNs in vitro [119]. To direct neural progenitors towards an LGE fate, various protocols employ fine-tuned SHH signalling activation and WNT signalling inhibition during neural induction to minimise the production of MGE or other lineages [120,121,122,123,124] (Figure 4). Arber et al. subsequently developed the Activin A strategy, activating the ALK4/5-SMAD2/3 signalling pathway that induces the transcription factors associated with LGE fate (Figure 4) without upregulating MGE markers, resulting in the generation of 20–40% of neurons expressing GABAergic and MSN markers such as GABA, GAD65/67, BCL11B, and PPP1R1B (Figure 4) [125]. Importantly, single-cell RNA-sequencing has confirmed that PSC-derived MSNs recapitulate the transcriptomic characteristics of human foetal MSNs and encompass both DRD1- and DRD2-expressing MSN subtypes [126].

#### 5.2.3. Differentiating Midbrain Dopaminergic Neurons

The development of mDA neurons relies on SHH signalling for the ventral identity specification of the midbrain floor plate [127,128], while WNT1 and FGF8 signalling from the midbrain–hindbrain boundary play crucial roles in specification and neurogenesis [129,130,131,132]. Highly efficient protocols have been developed to differentiate mDA neurons in vitro by finely tuning the timing, dose, and combination of SHH, WNT, and FGF8 signalling [133] (Figure 4). More recent work further refined the timing and dose of WNT signalling activation using small molecules such as CHIR99021, a GSK3ß inhibitor [134,135,136,137], or by inhibiting FGF/ERK signalling with PD0325901 upon epiblast exit [138], with this approach enabling both the differentiation of functional mDA neurons and mDA neuronal subtypes expressing CALB1 and GIRK2 [137,139].

#### 5.2.4. Differentiating Cerebellar Neurons

Cerebellar neurons encompass glutamatergic and GABAergic neurons distributed across different layers of the cerebellum [140]. GABAergic neurons include Purkinje cells and multiple interneuron types in the molecular layer, as well as unipolar brush cells in the internal granule layer [141]. Glutamatergic neurons, such as deep cerebellar nuclear projection neurons and granule cells in the internal granule layer, originate from the rhombic lip of the dorsal midbrain [141]. The intricate molecular mechanisms governing cerebellar development have been comprehensively reviewed elsewhere [141,142]. Based on these developmental principles, most differentiation paradigms have employed FGF8 and WNT signalling activation, SHH signalling inhibition, and transient FGF2 treatment in neural progenitors [143] (Figure 4). Using this paradigm, early studies patterned embryoid body-derived neural progenitors into cerebellar neuronal precursors resulting in the generation of Purkinje and granule cell-like neurons, albeit with a relatively low efficiency [144,145,146,147]. A major challenge has been the poor long-term survival of differentiated cerebellar neurons, partially addressed by co-culture with rodent granular progenitors or cerebellar slices [143,148,149]. An alternative approach has involved self-organising three-dimensional cell culture conditions, resulting in the generation of cerebellar organoids containing various cerebellar neuronal subtypes from both mouse and human PSCs [150,151,152]. More recently, Behesti et al. optimised a monolayer differentiation protocol using transwells to efficiently differentiate granule cells from human PSCs [153], while Hua et al. demonstrated that the late-stage activation of SHH signalling improved the efficiency of cerebellar neuronal differentiation [154]. However, early SHH signalling inhibition during cerebellar organoid differentiation has also been shown to be crucial in cerebellar lineage specification in vitro, although extended treatment with the SHH signalling inhibitor, cyclopamine, failed to improve the overall efficiency [155]. Despite these advancements, differentiating cerebellar neurons from human PSCs remains challenging, particularly in generating functional Purkinje cells due to their complex morphology and electrophysiological properties [148,156].

## 6. PSC-Derived Neuronal Models of MD

Although PSC-derived neuronal models have been widely used in the investigation of multiple neurological disorders [97], their application in studying MD remains limited. Kutschenko et al. focused on the differentiation of two patient-derived *SGCE* mutation-positive iPSCs (c.298T>G, p.W100G mutation and c.304C>T, p.R102X mutation) towards striatal MSNs with comparison to two control iPSC lines from age- and sex-matched unaffected individuals (Figure 5A) [9]. Morphologically, the *SGCE* mutation carrying MSNs exhibited fewer GABA-positive synaptic boutons [9], while, functionally, the mutant MSNs displayed a higher basal Ca^2+^ content and less active spontaneous Ca^2+^ transients but a more pronounced response to glycine and acetylcholine stimulation, consistent with the reported observed clinical benefit of anti-cholinergic therapy in some clinical settings for those diagnosed with dystonia. Interestingly, using different acetylcholine receptor antagonists, they also observed that acetylcholine-evoked Ca^2+^ responses in mutant MSNs were mediated more by muscarinic than nicotinic acetylcholine receptors [9]. Electrophysiological analyses using whole-cell patch clamp recordings also found the *SGCE*-mutation positive lines to have higher amplitudes of evoked action potentials and miniature post-synaptic currents, compared to controls, with the collective findings suggesting that mutation-positive MSNs may be more excitable. However, this study examined MSNs as a collective group, rather than the individual analysis of DRD1 and DRD2subtypes; thus, it remains unclear how hyperexcitable striatal MSNs impact the basal ganglia–thalamo-cortical circuit. Secondly, the MSN differentiation protocol was relatively inefficient, producing <20% PPP1R1B^+^ and BCL11B^+^ neurons of all DAPI^+^ nuclei, compared to almost 40% in other studies [126,157,158], with a potential impact on both calcium imaging and whole-cell patch clamp assays due to the difficulties in identifying authentic MSNs cells. Thirdly, the control and mutant cell lines were of different genetic backgrounds, potentially reducing the power of the study to identify a difference between the cell lines [159].

Sperandeo et al. investigated the effects of *SGCE* mutations on cortical glutamatergic neurons using iPSCs from three patients with *SGCE* mutations (one with c.771_772delAT, p.C258X mutation and two lines carrying c.622G>A, p.G221A mutation) and their CRISPR-Cas9 corrected isogenic control lines, coupled with an *SGCE* knock-out human ESC line from the iCas9 line (Figure 5B) [12]. These cell lines were comparable in their differentiation towards cortical glutamatergic neurons, with efficiencies of >50% and consisting of both TBR1^+^ upper and BCL11B^+^ deep layer neurons. Bulk RNA-sequencing found that the *SGCE* knock-out cortical glutamatergic neurons had a higher expression of genes related to axon projection and synaptic organisation and function but a lower expression of genes related to protein transport and cell adhesion and migration. Consistent with these transcriptomic alterations, the mutant cortical glutamatergic neurons exhibited longer neurites with more complex branching structures, longer axon initial segments, and significantly different synaptic organisation compared to their isogenic controls, consistent with the implicated changes in synaptic plasticity in the aetiology of dystonia. Functionally, the *SGCE* mutation-carrying neurons were more electrophysiologically active and excitable at both single-cell and network levels, with more frequent, but smaller, calcium transients with Fluo-4-based calcium imaging, potentially consistent with the overall hyperexcitable clinical phenotype of MD coupled with the hypothesised disruption to the basal ganglia–cerebello-thalamo-cortical circuits. However, the functional comparison of mutant neurons and their isogenic controls did not distinguish between different glutamatergic subtypes, which may be of relevance to both MD and the wider spectrum of dystonic disorders [160].

## 7. Future Perspectives

Existing in vivo neurophysiological and imaging studies have identified the involvement of the basal ganglia–cerebello-thalamo-cortical circuit in the pathogenesis of MD, with rodent and PSC-derived neuronal models implicating common cellular mechanisms including disruption to cytoskeletal organisation impacting neurite development and potentially synaptic organization [58]. In addition to structural changes, *SGCE* mutations likely impact synaptic function, possibly influencing the pre-synaptic release of neurotransmitters and the post-synaptic response [58]. These mechanisms align with the hypothesized role of ε-sarcoglycan and the DGC complex in synaptic organization and function [1]; however, the studies to date are few, and several key factors require further investigation (Figure 5C).

Firstly, the role of different neuronal subtypes across the distinct brain regions requires detailed characterisation, with work to date limited to MSNs and cortical glutamatergic neurons [9,12]. Of particular importance are cortical GABAergic interneurons and cerebellar Purkinje cells, where ε-sarcoglycan and the DGC potentially play a role in the formation and function of GABAergic synapses and where disruption to GABAergic neurotransmission may influence overall excitability in the pathways [1,41]. Moreover, neurophysiological studies suggest that reduced intracortical inhibition may contribute to the observed dystonia, and abnormal cerebellar function contributes to the myoclonus [1,72]. In addition, cortical GABAergic interneurons have been hypothesised to play an important role in the pathogenesis of several psychiatric disorders, providing a potential cellular basis for the psychiatric co-morbidity observed in MD [161].

Secondly, it is necessary to model the multiple interactions involved in the neuronal networks that contribute to dystonia, with these having been demonstrated across rodent models of dystonia [58,76,77]. To study these network mechanisms, heterogeneous monolayer PSC-derived neuronal models are inherently inadequate due to a lack of the complex environmental cues required for development and function, the unspecified identity of the by-product population, and the random formation of synaptic connections. To overcome some of these difficulties, a defined two-dimensional direct and indirect co-culture of neurons or three-dimensional assembloid strategies can be used to create complex neuronal networks composed of multiple types of neurons in vitro [162,163].

Thirdly, the molecular and cellular mechanisms by which *SGCE* mutations result in neuronal structural and functional changes require further investigation. Previous research has highlighted aspects of the biological processes and cellular components potentially affected by *SGCE* mutations, including synaptic structure and function and axonal projection [58]. However, the molecular machinery and pathways underlying these changes remain unclear. Part of the difficulty stems from the elusive interactome of ε-sarcoglycan. Although previous work found that ε-sarcoglycan interacts with other sarcoglycan proteins and dystrophin Dp71, the co-immunoprecipitation strategy used may have led to the dissociation of other binding partner proteins and the proteins that are transiently associated with ε-sarcoglycan or transported by mechanisms involving ε-sarcoglycan [40]. Additionally, transcriptomic approaches, while unbiased and high-throughput, do not always reflect changes at the protein level, leaving uncertainties around the quantitative and spatial proteomic alterations due to *SGCE* mutations and their downstream cellular effects.

Finally, although the benefits of PSC-derived cellular models have been highlighted above, limitations do remain in their ability to recapitulate human disease. Firstly, and particularly in 2D models, PSC-derived cellular models do not fully recapitulate the diverse cell populations or the complex and dynamic cell-to-cell interactions and extracellular environment that exist in vivo, limiting their recapitulation of the phenotypes and properties observed in vivo [164]. To this end, complex 3D models, such as organoids and assembloids, coupled with microfluidic devices, have been developed to allow for more representative platforms, although further work is needed to improve their reproducibility and utility in disease modelling [165]. Moreover, PSC-derived cellular models are inherently heterogeneous. Single-cell RNA-sequencing studies have identified unwanted and unexpected cell types in PSC-derived cell cultures and organoids, while cells at the same time point of differentiation may demonstrate distinct levels of maturation [166,167,168]. Additionally, as discussed above, the differentiation efficiency and maturity of PSC-derived neuronal subtypes remain limited and variable, requiring ongoing development of more efficient differentiation protocols, coupled with the comprehensive characterization of the resultant neurons.

## 8. Conclusions

In conclusion, MD is a complex network disorder whose underlying pathophysiological mechanisms remain largely unexplored. PSC-derived neuronal models offer a valuable platform to further investigate MD with the aim of identifying and validating novel therapeutic targets and strategies. These will likely need to extend beyond the monolayer cultures of individual neuronal types, extending to more complex three-dimensional structures that will also likely provide insight into the wider spectrum of dystonia disorders.

## Figures and Tables

**Figure 1 cells-13-01520-f001:**
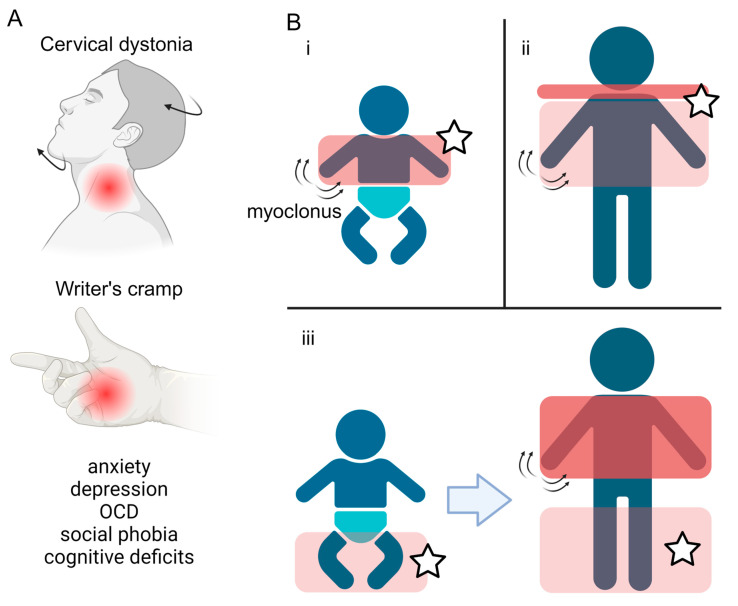
Symptoms and disease progression of Myoclonus Dystonia. (**A**) Overview of the motor and non-motor features typically observed in Myoclonus dystonia, including dystonia of the neck (cervical dystonia) and upper limbs (writer’s cramp), coupled with a spectrum of psychiatric co-morbidities. (**B**) Three main patterns of motor symptom evolution (**i**–**iii**) in Myoclonus dystonia patients. The red boxes highlight the body regions affected, with the darker colour representing more severe symptoms. Stars label areas where dystonia manifests. Created with BioRender.com.

**Figure 2 cells-13-01520-f002:**
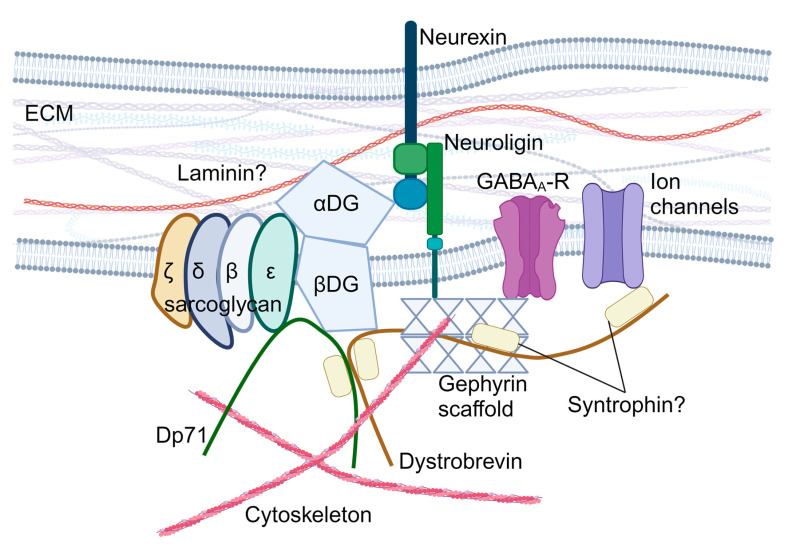
Hypothesised role of ε-sarcoglycan in the brain dystroglycan complex. αDG: α-dystroglycan; βDG: β-dystroglycan; DGC: dystroglycan complex; Dp71: dystrophin protein 71; ECM: extracellular matrix; and GABA_A_-R: GABA_A_ receptors. Created with BioRender.com.

**Figure 3 cells-13-01520-f003:**
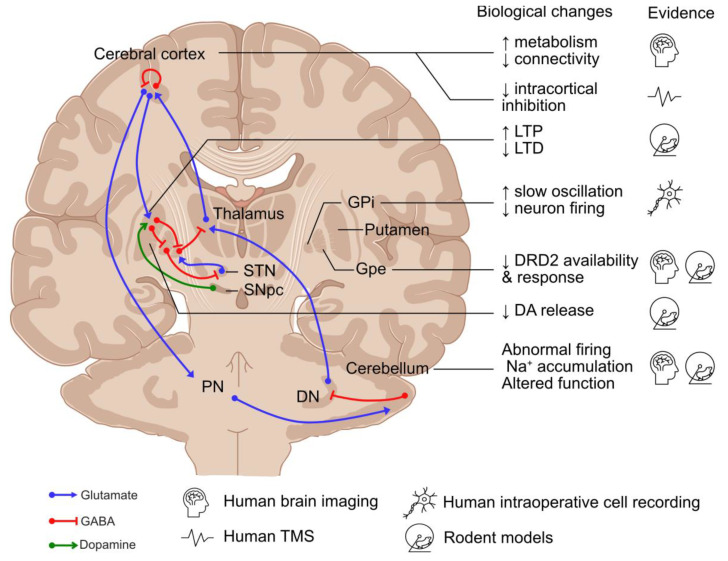
Neuronal networks involved in the pathogenesis of dystonia and existing evidence. DA: dopamine; DN: deep cerebellar nuclei; DRD2: dopamine receptor D2; GPe: globus pallidus externus; GPi: globus pallidus internus; LTD: long-term depression; LTP: long-term potentiation; PN: pontine neurons; SNpc: substantia nigra pars compacta; STN: subthalamic nucleus; and TMS: transcranial magnetic stimulation. Created using an icon resource from BioRender.com.

**Figure 4 cells-13-01520-f004:**
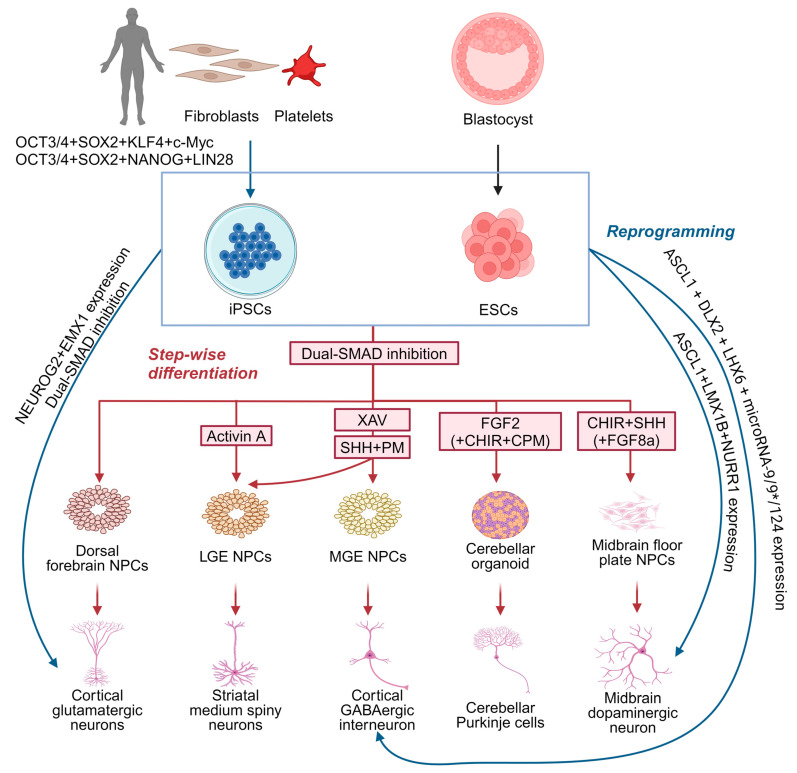
Strategies for generating distinct neuronal subtypes from pluripotent stem cells for dystonia disease modelling. Neurons can be generated from pluripotent stem cells using two strategies: direct reprogramming (blue) and step-wise differentiation (red). CHIR: CHIR99021, WNT signalling activator by inhibiting glycogen synthase kinase-3 beta; CPM: cyclopamine, Sonic hedgehog signalling inhibitor by inhibiting Smoothened protein; FGF2: fibroblast growth factor 2; FGF8a: fibroblast growth factor 8a; PM: purmorphamine, Sonic hedgehog signalling activator by activating Smoothened protein; SHH: Sonic hedgehog, activating SHH signalling; and XAV: XAV939, WNT signalling inhibitor by inhibiting tankyrase 1 and 2. Created with BioRender.com.

**Figure 5 cells-13-01520-f005:**
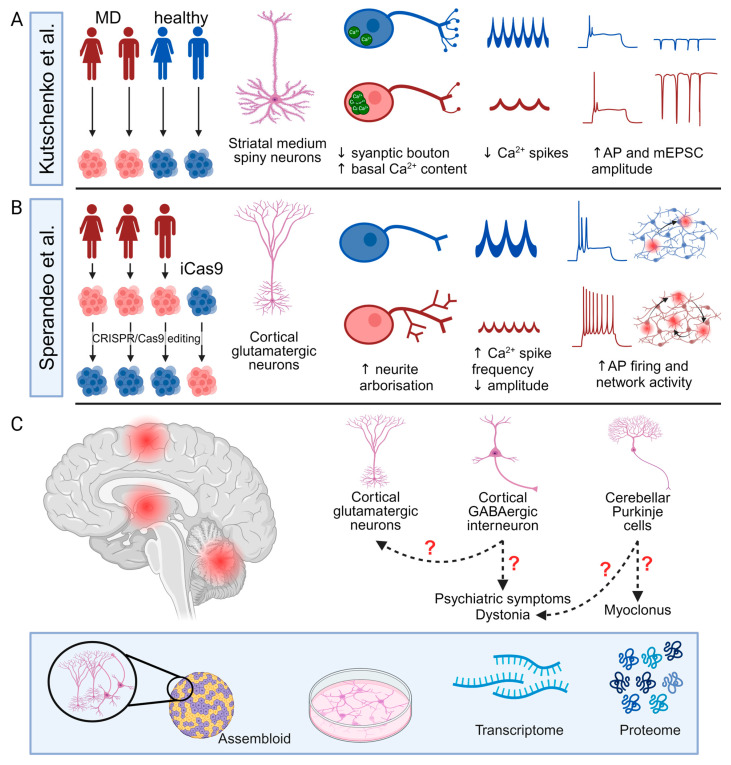
Current progress of pluripotent stem cell-derived cellular models for Myoclonus dystonia and future directions. (**A**,**B**) Summary of the study design and main findings of two studies of *SGCE* mutation-positive Myoclonus dystonia (MD) using pluripotent stem cell-derived neurons including patient-derived lines (red) and control lines (blue). (**C**) Potential future directions of MD research using PSC-derived three-dimensional assembloids and two-dimensional cellular models. AP: action potential; and mEPSC: miniature excitatory post-synaptic current. Created with BioRender.com.

**Table 1 cells-13-01520-t001:** Symptomatic oral medical therapies for the treatment of Myoclonus Dystonia.

Strategy	Drug	Mechanism	Reference
Pharmacological treatment	Zonisamide	Inhibit Na^+^ and T-type Ca^2+^ channel Modulate GABAergic, glutamatergic, and dopaminergic transmission	[19]
	Carbamazepine	Voltage-dependent Na^+^ channel antagonist Increase dopaminergic and serotonergic transmission	[20]
	Benzodiazepines	Enhance GABAergic transmission	[21]
	Trihexyphenidyl	Inhibit cholinergic transmission	[22]
	Tetrabenazine	Inhibit vesicular monoamine transporter 2	[23]
	Levodopa	Monoamine precursor	[24]
	L-5-hydroxytryptophan	Serotonin precursor	[25]
	Sodium Oxybate	Enhance GABA-B transmission	[26]
	Botulinum toxin	Unknown	[27]

## Data Availability

Data sharing is not applicable. No new data were created or analysed in this study.

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
