# Peer review of "Deciphering the Pathophysiological Mechanisms Underpinning Myoclonus Dystonia Using Pluripotent Stem Cell-Derived Cellular Models"

_cells, 2024, doi:10.3390/cells13181520_

Round 1

Reviewer 1 Report

Comments and Suggestions for Authors

The paper by Zongze and coworkers provides a detailed and well-structured overview of Myoclonus Dystonia (MD), specifically focusing on the role of pluripotent stem cell-derived cellular models. The manuscript emphasizes the relevant potential of pluripotent stem cells (PSCs) in modeling MD and identifying novel therapeutic targets. 

Pepr is clear and well written, in my opinion only few minor points require revision

The manuscript discusses the differentiation of various neuronal subtypes, and the focus is primarily on striatal medium spiny neurons and cortical glutamatergic neurons. However other potentially relevant subtypes, such as cortical GABAergic interneurons and cerebellar Purkinje cells, could be more discussed and explored in depth.

The manuscript emphasizes in vitro models, but there is a lack of discussion on how findings from these models correlate with in vivo data. This could improve the relevance of the in vitro findings to the actual pathophysiology of Myoclonus Dystonia.

The authors should provide a more detailed discussion on the limitations of current PSC-derived models and suggest possible solutions or alternative methods

Comments on the Quality of English Language

The English is clear and effective but could benefit from minor revisions to enhance clarity.

Author Response

Many thanks for your ongoing consideration of the above manuscript. Please find below our response to each of the Reviewers’ comments.

Comment 1: The manuscript discusses the differentiation of various neuronal subtypes, and the focus is primarily on striatal medium spiny neurons and cortical glutamatergic neurons. However other potentially relevant subtypes, such as cortical GABAergic interneurons and cerebellar Purkinje cells, could be more discussed and explored in depth.

Response 1: Thank you for highlighting the relevance of cortical GABAergic interneurons and cerebellar Purkinje cells. Section 5.2.1 (page 9-10, line 273-297) and section 5.2.4 (page 11-15, line 325-351) discuss the differentiation of cortical GABAergic interneurons and cerebellar Purkinje cells, respectively. Unfortunately, for these two neuronal subtypes, current differentiation strategies remain limited. However, we have expanded on both areas in the discussion to provide further detail for the reviewer (page 10, line 289-290, 293-294, and page 10-11, line 345-348).

Comment 2: The manuscript emphasizes in vitro models, but there is a lack of discussion on how findings from these models correlate with in vivo data. This could improve the relevance of the in vitro findings to the actual pathophysiology of Myoclonus Dystonia.

Response 2: Many thanks for raising this point. In the revised section 6, we have further elaborated how in vitro findings correlate with in vivo findings (page 11, line 362-363 and 371-372 for MSNs, and line 391-392 and 395-396 for cortical glutamatergic neurons).

Comment 3: The authors should provide a more detailed discussion on the limitations of current PSC-derived models and suggest possible solutions or alternative methods

Response 3: We have added a new paragraph in section 7 (page 14-15, line 454-459), discussing the limitations of PSC-derived cellular models and possible solutions.

Reviewer 2 Report

Comments and Suggestions for Authors

Dystonia are a genetically heterogeneous group of movement disorders combined with psychiatric manifestations. The pathophysiological understanding is still limed and so are the therapeutic options. In this manuscript the authors review the state of the art in Myoclonus Dystonia due to heterozygous mutations in the epsilon-sarcoglycan gene, reconsider human in vivo electrophysiological studies, and animal models, and illustrate recent advances in induced pluripotent stem cells (iPSCs) and subsequent neuronal differentiation to study the molecular and cellular mechanisms of dystonia. The manuscript is well planned and it is informative.

Minor changes the authors must address:

1. Before introducing SGCE list the more common genetic forms of dystonia

2. A brief table to summarize main therapies in MD and related dystonia would make more appealing the manuscript

3. At the end of subheading #4 it seems useful to add a few comments on the why mouse models are not fully satisfactory in understanding the pathological mechanisms in dystonia or MD

4. Page 9, ln 302. Statement is correct but please also comment on work in PMID: 38411252

5. Limitations on the information in the iPSC derived neuronal types should be highlighted (including variability in the differentiation protocol used in the literature)

Comments on the Quality of English Language

minor editing

Author Response

Many thanks for your ongoing consideration of the above manuscript. Please find below our response to each of the Reviewers’ comments.

Comment 1: Before introducing SGCE list the more common genetic forms of dystonia

Response 1: Thank you for this suggestion, we have listed examples of Mendelian inherited forms of dystonia in the introduction (Page 1, lines 32-34).

Comment 2: A brief table to summarize main therapies in MD and related dystonia would make more appealing the manuscript

Response  2: We have included a table to summarise the main oral medical therapies that have been used for symptomatic treatment in Myoclonus Dystonia (Table 1, page 3-4, lines 81-82). In addition, we have discussed both oral medical therapies and surgical intervention in section 2 (page 2, lines 68-73).

Comment  3: At the end of subheading #4 it seems useful to add a few comments on the why mouse models are not fully satisfactory in understanding the pathological mechanisms in dystonia or MD

Response 3: Many thanks for this suggestion. We felt that a brief discussion of the limitations of dystonia murine models might be better placed at the beginning of section 5, providing a gateway to the discussion of the relative merits of PSC-derived models (section 5, page 7-8, line 208-215).

Comment  4: Page 9, ln 302. Statement is correct but please also comment on work in PMID: 38411252

Response 4: We have added this reference, discussing this work in the context of other PSC-derived cerebellar models in page 11-12, line 345-348.

Comment  5: Limitations on the information in the iPSC derived neuronal types should be highlighted (including variability in the differentiation protocol used in the literature)

Response 5: Many thanks once again. We have added an addition paragraph to the end of section 7 (page 14-15, line 454-459) discussing the limitations of PSC-derived cellular models in the context of disease understanding.